# Comprehensive somatic mutational analysis in glioblastoma: Implications for precision medicine approaches

**Parisa Azimi**[1], **Mina Karimpour**[2], **Taravat Yazdanian**[3], **Mehdi Totonchi**[4]*,
**Abolhassan Ahmadiani**[1]*

**1** Neuroscience Research Center, Shahid Beheshti University of Medical Sciences, Tehran, Iran,
**2** Department of Genetics, Faculty of Biological Sciences, Tarbiat Modares University, Tehran, Iran,
**3** School of Medicine, Capital Medical University, Beijing, China, **4** Department of Genetics, Reproductive Biomedicine Research Center, Royan Institute, Tehran, Iran

* totonchimehdi@gmail.com (MT); aahmadiani@yahoo.com (AA)

**Data Availability Statement:** The data that support the findings of this study were derived from the following resources available in the public domain:
1. CGGA raw WES fastq data (DataSet ID:

## Abstract

Glioblastoma multiforme (GBM), a malignant neoplasm originating from glial cells, remains challenging to treat despite the current standard treatment approach that involves maximal safe surgical resection, radiotherapy, and adjuvant temozolomide chemotherapy. This underscores the critical need to identify new molecular targets for improved therapeutic interventions. The current study aimed to explore the somatic mutations and potential therapeutic targets in GBM using somatic mutational information from four distinct GBM datasets including CGGA, TCGA, CPTAC and MAYO-PDX. The analysis included the evaluation of whole exome sequencing (WES) of GBM datasets, tumor mutation burden assessment, survival analysis, drug sensitivity prediction, and examination of domain-specific amino acid changes. The results identified the top ten commonly altered genes in the aforementioned GBM datasets and patients with mutations in *OBSCN* and *AHNAK2* alone or in combination had a more favorable overall survival (OS). Also, the study identified potential drug sensitivity patterns in GBM patients with mutations in *OBSCN* and *AHNAK2*, and evaluated the impact of amino acid changes in specific protein domains on the survival of GBM patients. These findings provide important insights into the genetic alterations and somatic interactions in GBM, which could have implications for the development of new therapeutic strategies for this aggressive malignancy.

## Introduction

Glioblastoma multiforme (GBM), a highly malignant type of central nervous system (CNS) tumor that originates from glial cells in the brain, is known for its aggressive nature and poor prognosis. It is the most common primary brain tumor in adults and accounts for approximately 14.5% of all CNS tumors [1]. The standard first-line treatment for GBM involves maximal safe surgical resection followed by radiotherapy and adjuvant temozolomide chemotherapy, resulting in a median overall survival (OS) of 12–15 months and a 5-year OS

WESeq_286, BIGD accession number: PRJCA001636) and expression data (DataSet ID: mRNAseq_693) from Chinese Glioma Genome Atlas database (http://www.cgga.org.cn/download. jsp) 2. TCGA-GBM somatic mutation, expression data, and clinical information from UCSC-Xena browser (https://xenabrowser.net) 3. CPTAC and MAYO-PDX glioblastoma somatic mutation and expression data from cBioportal for Cancer Genomics database (https://www.cbioportal.org).

**Funding:** The authors received no specific funding for this work.

**Competing interests:** The authors have declared that no competing interests exist.

**Abbreviations:** CNS, Central Nervous System; CGGA, Chinese Glioma Genome Atlas; CPTAC, Clinical Proteomic Tumor Analysis Consortium; CI, Confidence Interval; FDR, False Discovery Rate; FN3, Fibronectin type 3 domain; GBM, Glioblastoma Multiforme; RhoGEF, Guanine nucleotide exchange factor for Rho/Rac/Cdc42-like GTPases; IC50, half-maximal inhibitory concentrations; HR, Hazard ratio; IGF-IR, insulin growth factor receptor; Ig/IG, Immunoglobulin domain; I-set, Immunoglobulin I-set domain; Ig Semaphorin C, Immunoglobulin-like domain of semaphoring; MAYO-PDX, Mayo Clinic Brain Tumor Patient-Derived Xenograft National Resource; MB, MegaBase; Mut, Mutant; OS, Overall Survival; Pfam, Protein families; S-TKc, Serine/Threonine protein kinases; catalytic domain; SH3, Src homology 3; TCGA, The Cancer Genome Atlas; TMB, Tumor Mutation Burden; WES, whole exome sequencing; WT, Wild Type; WHO, World Health Organization.

rate of less than 5%. Despite this optimal treatment, the rate of local recurrence remains high, at approximately 90% [2, 3]. Therefore, it is essential to identify new molecular targets that are involved in cell growth and survival to develop more effective therapeutic approaches for the GBM [4]. The classification of GBM has been refined over several years through updates to the World Health Organization's (WHO) classification systems. The 2007 CNS WHO classification system categorized glial tumors based on their astrocytic phenotype, without taking into account molecular features [5]. However, the 2016 classification system incorporated molecular IDH mutational status, providing a more comprehensive and accurate understanding of the underlying biology of brain tumors [6]. More recently, the WHO 2021 CNS classification divides diffuse gliomas into adult and pediatric types, with adult types including astrocytoma IDH-mutant, oligodendroglioma IDH-mutant along with 1p/19-codeletion, and glioblastoma IDH-wildtype. Astrocytoma IDH-mutant is graded as CNS WHO II, III, or IV [7].

The complex genetic profile of GBM patients has been revealed through various omics studies such as the Chinese Glioma Genome Atlas (CGGA) and The Cancer Genome Atlas (TCGA) [8, 9]. These studies have highlighted frequent mutations in *PTEN*, *TP53*, *TERT*, *IDH1*, and *ATRX* genes, as well as *EGFR* gene amplification and 1p/19q co-deletions [10]. The availability of genetic information is enhancing the precision of GBM diagnosis and treatment which ultimately results in better patient outcomes. In this regard, molecular targeted therapy has shown promise in the treatment of several types of cancer including GBM. Currently, the only FDA-approved targeted therapy for recurrent GBM patients is bevacizumab, a monoclonal antibody that targets VEGF and blocks the formation and maintenance of tumor blood vessels [11]. In addition, the development of prognostic GBM biomarkers often relies on surgically obtained tumor samples. However, bias due to varying surgeon selection criteria affects the accuracy of future analyses [12]. Thus, by integrating multiple datasets and considering larger sample sizes, it appears that the current limitations could potentially be mitigated.

In this study, we aimed to analyze somatic mutation data acquired from whole exome sequencing (WES) of GBM patients from different independent datasets to identify frequently occurring genetic alterations that are significantly associated with GBM patients' survival. Our analysis revealed several commonly mutated genes, which could potentially indicate survival outcomes in GBM patients.

## Methods

### Data sources

We used the somatic mutational information derived from four distinct GBM datasets, namely CGGA, TCGA, Clinical Proteomic Tumor Analysis Consortium (CPTAC), and Mayo Clinic Brain Tumor Patient-Derived Xenograft National Resource (MAYO-PDX). Raw WES fastq files and associated clinical information of 102 GBM patients were obtained from the CGGA database [8]. Somatic mutation data and clinical information of 461 GBM patients from the TCGA project (https://portal.gdc.cancer.gov/, accessed October 2022) were provided from the UCSC-Xena database. Furthermore, We obtained publicly available gene expression datasets for GBM patients from the CGGA (n = 249) and TCGA (n = 175) databases. We also acquired mutation annotation and expression data of 99 and 83 GBM samples from the CPTAC (https://portal.gdc.cancer.gov/) and MAYO-PDX [13] cohorts, respectively, through the cBioportal database.

### CGGA whole exome sequencing analysis

We analyzed CGGA-GBM WES data, which comprises paired-end fastq files of both tumor and blood samples from 102 patients. We performed several computational steps to extract

somatic mutational information. Firstly, we aligned the fastq files to the hg38 human reference genome using the BWA software. The resulting SAM file was converted to BAM files and sorted using samtools. Subsequently, we applied mark duplication using Picard to reduce the impact of sequencing artifacts. To identify somatic SNVs and INDELs, we utilized the GATK somatic short variant discovery workflow, specifically GATK-MuTect2, followed by filtration using GATK-FilterMutectCall. Finally, we performed an annotation of the identified variants using the ANNOVAR web server.

## Tumor mutation burden assessment

To assess the tumor mutation burden (TMB) levels, we quantified the number of coding, somatic base substitutions, and indel mutations per megabase (MB) within the targeted region. Specifically, we counted all base substitutions and indels within the coding region of targeted genes, excluding silent mutations that do not result in amino acid variations. The TMB was computed for each patient using the 'tmb' function within the Maftools R package.

## Survival analysis

To assess OS, we employed the Kaplan-Meier method and compared survival curves using the log-rank test. Furthermore, we conducted univariate Cox proportional hazard regression analysis to estimate the Hazard ratio (HR) and 95% confidence interval (CI) of specific genes using the survminer package in R. We considered statistical significance to be present when the p-value was $\leq 0.05$, for both the log-rank and the Cox proportional hazard regression tests.

## Drug sensitivity prediction

We used the pRRophetic package in R to predict drug response based on pharmacogenomics and gene expression data using a ridge regression model. We obtained half-maximal inhibitory concentrations (IC50) for 138 different drugs in CGGA and TCGA patients, and then conducted Kruskal-Wallis and Wilcoxon rank-sum tests to compare differences between GBM patients with *OBSCN* or *AHNAK2* mutations and those with wild -type genes. We also examined three different phenotypes based on *OBSCN* and *AHNAK2* statuses (Double WT, Single WT, and Double Mut). To correct for multiple testing, we adjusted the *p*-values using the Benjamini-Hochberg (BH) method, and considered a false discovery rate (FDR) of less than 0.05 to be statistically significant.

## Domain-specific amino acid changes

We examined the specific amino acid changes in the *OBSCN* and *AHNAK2* proteins. To visualize these changes, we used the 'lollipopPlot' function from the Maftools package in R. This function uses the protein families (Pfam) database to identify the domains of each protein affected by the amino acid changes. Following this, we evaluated the impact of the amino acid changes within each domain on the survival of GBM patients.

# Results

## Exploring somatic mutations in GBM datasets

We examined the somatic mutations in four distinct datasets of GBM patients including CGGA, TCGA, CPTAC, and MAYO-PDX. Fig 1 displays the top 20 most frequently altered genes and their frequencies in each GBM dataset. The most commonly mutated genes in the CGGA dataset were *TP53* (55%), *TTN* (37%), *IDH1* (29%), *ATRX* (25%), and *MUC16* (23%). Meanwhile, in the TCGA dataset, the most frequently mutated genes were *PTEN* (33%), *TP53*

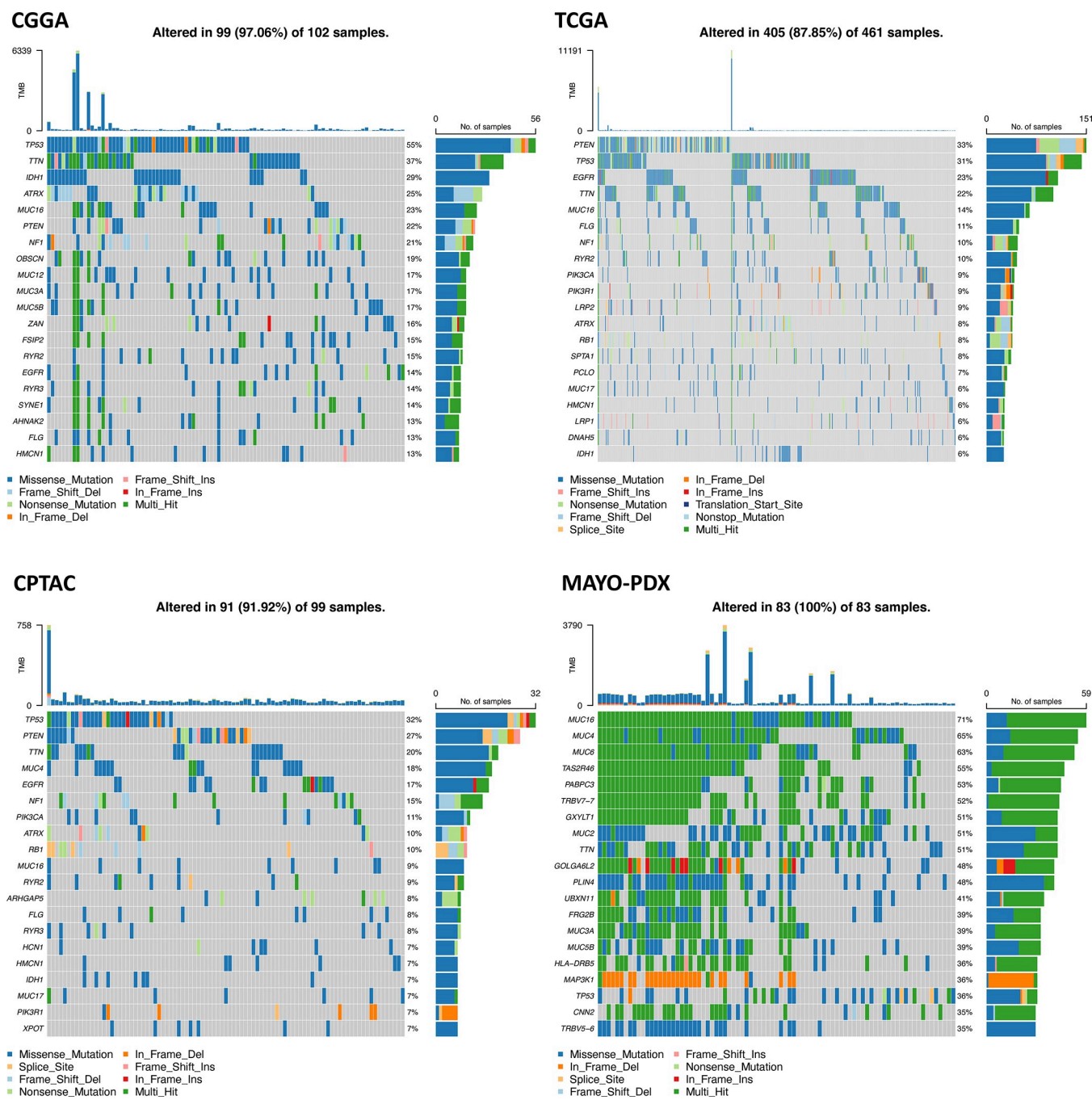

**Fig 1. Oncoplots of the top 20 somatic mutations in GBM across four independent datasets including CGGA, TCGA, CPTAC, and MAYO-PDX.** Each oncoplot comprises two bar plots, with the top bar plot showing the frequency of different variant classifications across the samples, and the right bar plot indicating the frequencies of various variant classifications in each gene.

(31%), *EGFR* (23%), *TTN* (22%), and *MUC16* (14%). The CPTAC dataset showed similar patterns to CGGA and TCGA, with *TP53* (32%), *PTEN* (27%), *TTN* (20%), *MUC4* (18%), and *EGFR* (17%) being the most highly mutated genes. However, the MAYO-PDX dataset revealed slightly different results, with *MUC16* (71%), *MUC4* (65%), *MUC6* (66%), *TAS2R46* (55%), *PABPC3* (53%), and *TRBV7-7* (52%) being the most frequently mutated genes. This may be

Table 1. Demographic information of four independent GBM datasets.

| Demographic categories | CGGA | TCGA | CPTAC | MAYO |
|---|---|---|---|---|
| **Age** | | | | |
| Over 60 | 23 | 197 | 46 | 41 |
| Under 60 | 74 | 165 | 53 | 40 |
| **Gender** | | | | |
| Female | 45 | 137 | 44 | 34 |
| Male | 52 | 225 | 55 | 47 |
| **IDH-status** | | | | |
| IDH1-Mut | 30 | 22 | 11 | 4 |
| IDH1-WT | 67 | 340 | 88 | 77 |
| **Subtype** | | | | |
| Primary | 53 | *NA* | 30 | *NA* |
| Recurrent | 44 | *NA* | 39 | *NA* |
| **Radiotherapy** | | | | |
| Treated | 81 | 114 | *NA* | 27 |
| Un-treated | 11 | 248 | *NA* | 54 |
| **Chemotherapy** | | | | |
| Treated | 80 | 107 | *NA* | 25 |
| Un-treated | 8 | 255 | *NA* | 55 |
| **X1p19q_codeletion** | | | | |
| Codel | 6 | 0 | *NA* | *NA* |
| Non-codel | 55 | 351 | *NA* | *NA* |
| **MGMTp_methylation** | | | | |
| methylated | 41 | 136 | *NA* | *NA* |
| un-methylated | 26 | 168 | *NA* | *NA* |

*NA*; Not available.

because WES was performed on patient-derived xenograft samples. Nonetheless, a comparison of somatic alterations in 24 matched patient tumors and derivative PDX in this project showed significant concordance between patient and PDX, so we decided to include this dataset in subsequent analysis [13]https://paperpile.com/c/uoHlvz/QNP4. Table 1 presents a comprehensive overview of the demographic information for the patients included in the aforementioned GBM datasets.

## Common somatic mutations and interactions in GBM datasets

We explored genes that had mutations in at least one sample from each of the four evaluated datasets of GBM (S1 Table in S1 File). Since, our study aimed to identify the most commonly mutated genes in GBM patients, we examined the top 100 most frequently mutated genes and identified ten genes that were commonly altered across all datasets, including *PTEN, TP53, TTN, MUC16, FLG, PCLO, MUC17, HMCN1, AHNAK2*, and *OBSCN* (Fig 2A). Additionally, we investigated the tumor mutational burden (TMB) in each cohort and found that the median values of TMB were 1.72/MB, 0.88/MB, 0.84/MB, and 2.76/MB for CGGA, TCGA, CPTAC, and MAYO-PDX, respectively. Moreover, our analysis revealed that patients with mutations in *AHNAK2, FLG, HMCN1, MUC16, MUC17, OBSCN, PCLO*, and *PTEN* displayed significantly higher TMB in all four cohorts (Fig 2B). In the next step, we integrated the mutational information of these four datasets and investigated the somatic interactions between ten

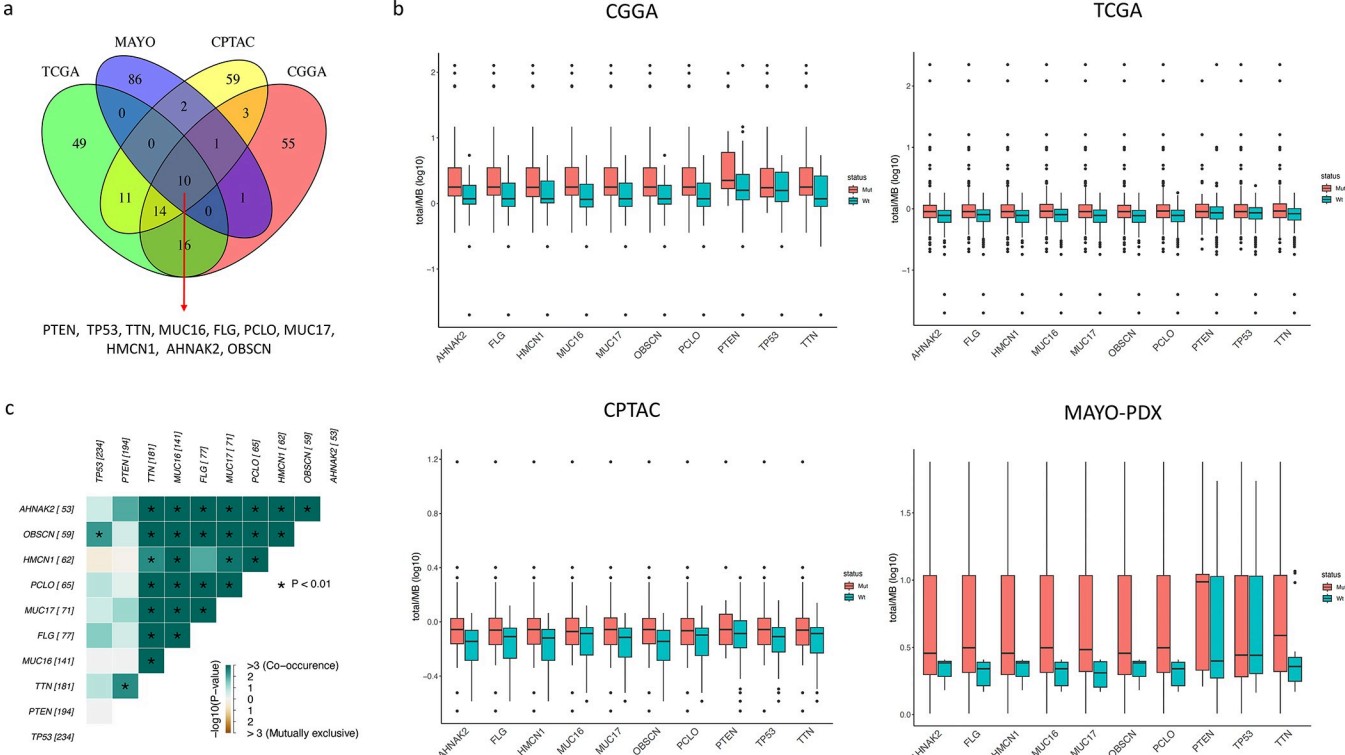

**Fig 2. Common somatic mutations were observed in four independent GBM datasets including CGGA, TCGA, CPTAC, and MAYO-PDX.** a) A Venn-diagram depicting the overlap and unique somatic mutations detected in the four GBM datasets. b) TMB box plots for the 10 commonly mutated genes in GBM datasets, compared to each gene's wild-type samples. c) Somatic interactions are shown as co-occurrence or mutually exclusive events among the 10 commonly mutated genes in GBM.

commonly mutated genes and found significant correlations between most of the possible pairs. Notably, *MUC17-MUC16*, *MUC17-PCLO*, *PCLO-MUC16*, *TTN-PCLO*, and *OBSCN-TTN* interactions showed the strongest correlations (*q*-value < 0.00001; Fig 2C, S2 Table in S1 File). These findings provide important insights into the genetic alterations and somatic interactions in GBM, which could have implications for the development of new therapeutic strategies, particularly those that are tailored to the individual patient's genetic profile.

## Correlation between commonly mutated genes and OS

We carried out a Kaplan-Meier survival analysis to explore the potential association between mutations in ten commonly mutated genes and overall survival in GBM patients of integrated datasets. Our results revealed that patients with mutations in *OBSCN* or *AHNAK2* had a more favorable OS (Fig 3A and 3B). Furthermore, we employed univariate Cox regression analysis to determine the hazard ratios (HRs) of *OBSCN* and *AHNAK2*, which were found to be 1.4 (95% CI: 1–2) and 1.5 (95% CI: 1.1–2.2), respectively (Fig 3C). We also included supplementary Fig 1, which presents the Kaplan-Meier survival analysis for *PTEN*, *TP53*, *TTN*, *MUC16*, *FLG*, *PCLO*, *MUC17*, and *HMCN1*, all of which were not statistically significant between mutant (Mut) and wild-type (WT) groups.

As in previous sections, we demonstrated that *OBSCN* and *AHNAK2* had a predominant somatic interaction (*q*-value = 2.05e-6; Fig 2C, S2 Table in S1 File). Hence, we suggested that the mutational status of *OBSCN* and *AHNAK2* together might be closely related to the survival outcome and underlying biological mechanisms of GBM patients. In this regard, we classified

patients into three phenotypes based on their *OBSCN* and *AHNAK2* mutational statuses: Double-Mut, Single-Mut, and Double-WT. We found that the Double-WT, Single-Mut, and Double-Mut phenotypes were associated with shorter OS, intermediate OS, and longer OS, respectively (p = 0.018; Fig 3D). The HR of the *OBSCN-AHNAK2* mutation phenotypes was 0.71 (p < 0.05; Fig 3D). Furthermore, we noted that patients with the Double-Mut phenotype had the highest TMB in the integrated dataset relative to the other two phenotypes (Fig 3E).

We also conducted Fisher's exact test to determine whether there were significant differences in age, gender, or IDH1 status of GBM patients among the different phenotypes based on *OBSCN or/and AHNAK2* statuses. Our analysis indicated that there were no statistically significant differences in age, gender, or *IDH1* status of *AHNAK2* mutant and wild-type GBM patients (Fig 3F). However, patients with mutations in the *OBSCN* gene were more likely to be under 60 years old (p = 0.018; Fig 3G). Furthermore, our analysis of *OBSCN* and *AHNAK2* statuses in combination revealed that age was a clinically significant factor that shows a significant difference in Double-Mut patients compared to the other two phenotypes (Fig 3H).

We analyzed to assess the impact of *OBSCN* or/and *AHNAK2* expression levels on the survival rates of GBM patients. Our results indicated that high levels of *AHNAK2* expression were significantly associated with poor survival rates in GBM patients from the CGGA and TCGA datasets, when compared to those with low *AHNAK2* expression (S2 Fig in S1 File). However, we did not observe any significant differences in survival rates based on the expression levels of *OBSCN* alone or in combination with *AHNAK2* (S2 Fig in S1 File).

## Exploring potential drug sensitivity patterns in GBM patients with *OBSCN* and *AHNAK2* mutations

We analyzed 138 chemotherapeutic and targeted agents in GBM patients to determine potential drugs that exhibit preferential sensitivity to mutations in either *OBSCN* or/and *AHNAK2*. Our findings indicate that patients with mutations in *OBSCN* are significantly more sensitive to eight potential drugs, namely Thapsigargin, BMS.754807, BAY.61.3606, OSI.906 (Linsitinib), Cytarabine, Embelin, IPA.3, and AZD7762 (Kruskal-Wallis and Wilcoxon rank-sum test $q$-values < 0.05; Fig 4A). The potential targets of each drug are presented in Fig 2A. No drug exhibited a statistically significant lower IC50 in *AHNAK2*-mutated GBM patients. However, our analysis of the sensitivity of drugs based on both *OBSCN* and *AHNAK2* statuses suggests that OSI.906 and BMS.754807 have the potential to sensitize patients with double mutations in *OBSCN* and *AHNAK2* compared to single mutant and double wild-type phenotypes (Fig 4C). Remarkably, both drugs are inhibitors of the insulin growth factor receptor (IGF-IR), which is identified as independent prognostic factors associated with shorter survival and a less favorable response to temozolomide in GBM patients [14]. It is worth noting that OSI.906 and BMS.754807 may also exhibit potential effects in GBM patients with only *AHNAK2* mutations. However, the mutant group did not achieve statistical significance after multiple testing corrections (Kruskal-Wallis and Wilcoxon rank-sum test $p$-values < 0.05).

## Analysis of protein level mutations of *OBSCN* and *AHNAK2* in GBM patients

The mutational impact on protein levels of OBSCN and AHNAK2 was analyzed using lollipop plots generated by Maftools, which highlight amino acid variations and indicate mutations in different protein domains using various colors (Fig 5A and 5B). The Pfam database suggested several protein domains for the OBSCN protein, including the Immunoglobulin domain (Ig/IG), Immunoglobulin I-set domain (I-set), Immunoglobulin-like domain of semaphoring (Ig Semaphorin C), Fibronectin type 3 domain (FN3), Serine/Threonine protein kinases, catalytic

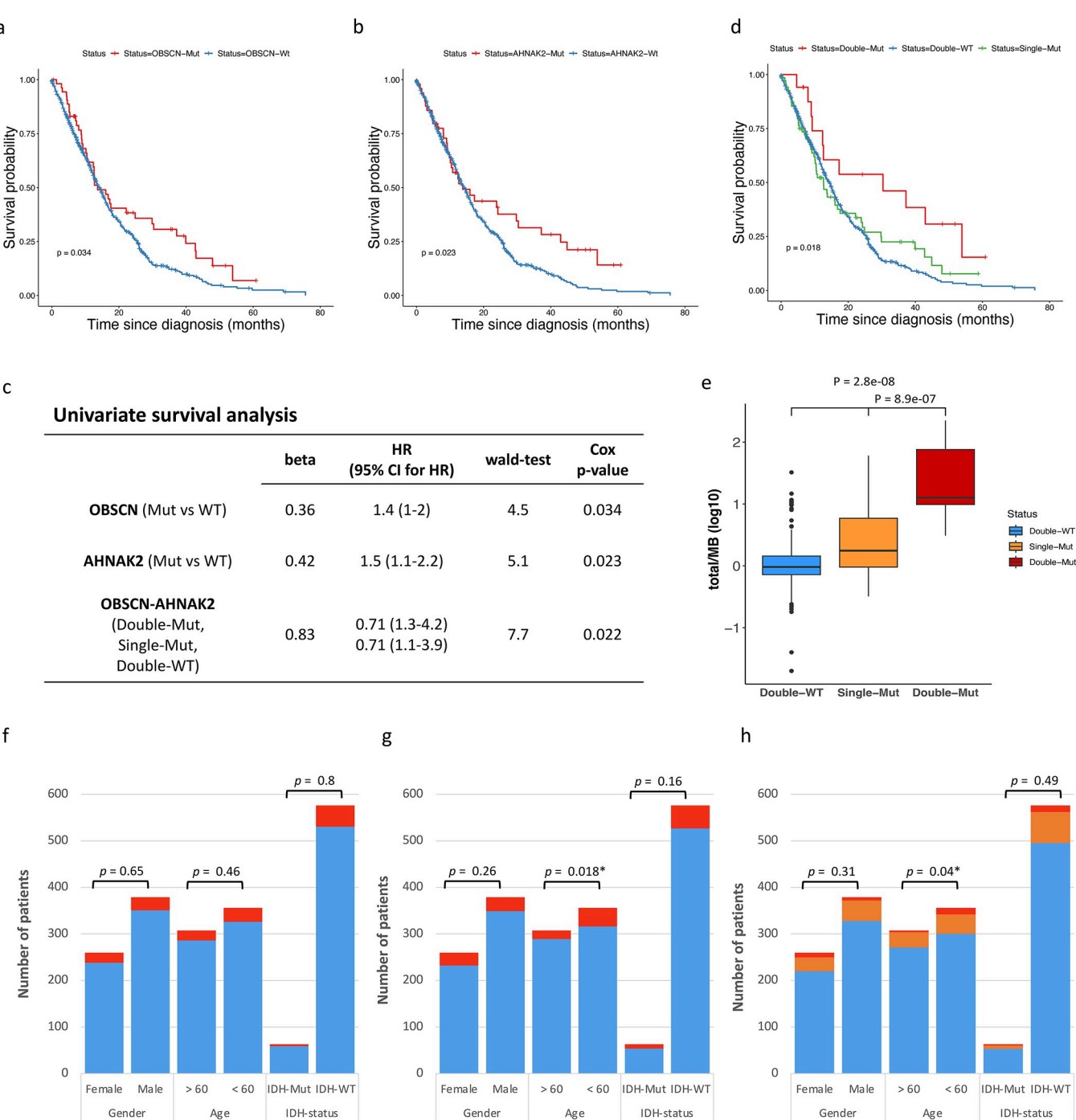

**Fig 3. OS Analysis of *OBSCN* and *AHNAK2* mutations in integrated GBM datasets and their association with clinical factors.** a) The Kaplan-Meier curve for OS analysis in *OBSCN* mutant and wild-type (Wt) GBM patients. b) The Kaplan-Meier curve for OS analysis in *AHNAK2* mutant and wild-type (WT) GBM patients c) Univariate Cox regression analysis for *OBSCN* and/or *AHNAK2* mutations versus wild-type GBM patients. d) The Kaplan-Meier curve for GBM patients with *OBSCN* and *AHNAK2* double mutations (Double-Mut) compared to Single-Mut and Double-WT phenotypes. e) TMB levels in Double-WT, Single-Mut and Double-Mut phenotypes based on *OBSCN* and *AHNAK2* statuses. f) The number of GBM patients with *AHNAK2* mutation or wild-type status concerning age, gender, and *IDH1* mutational status. g) The number of GBM patients with *OBSCN* mutation or wild-type status concerning age, gender, and IDH1 status. h) The number of GBM patients with *OBSCN* and *AHNAK2* Double-Mut, Single-Mut, and wild-type phenotypes concerning age, gender, and IDH1 status.

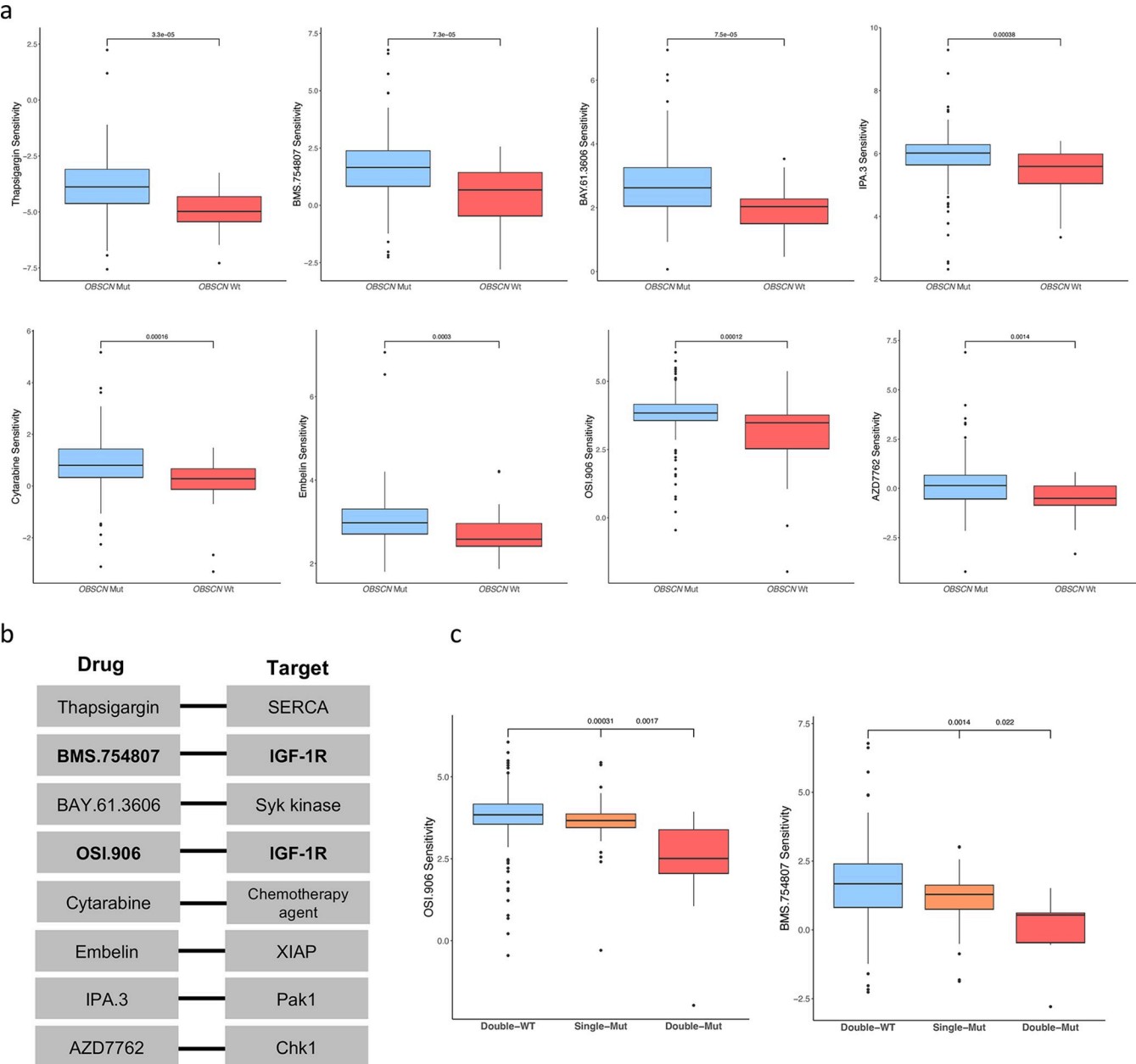

**Fig 4. Prediction of drug sensitivity using gene expression data with the pRophetic R package.** (a) Eight potential targeted and chemotherapeutic drugs significantly sensitize *OBSCN* mutants compared to *OBSCN* wild-type GBM patients. (b) The associated gene targets for drugs suggested for *OBSCN* mutations. (c) The effects of OSI.906 and BMS.754807, which are inhibitors of IGF-1R, on GBM patients with *OBSCN* and *AHNAK2* Double-Mut, Single-Mut, and wild-type phenotypes.

domain (S-TKc) and Guanine nucleotide exchange factor for Rho/Rac/Cdc42-like GTPases (RhoGEF) (S3 Table in S1 File). We investigated whether mutations in specific protein domains of OBSCN would affect the survival of GBM patients. The Kaplan-Meier curve indicated that mutations in different domains of the OBSCN protein resulted in significantly different survival statuses (Fig 5C). Furthermore, GBM patients with mutations in the Immunoglobulin domain had significantly better survival compared to patients with mutations in other Ig domains of the protein (Fig 5D). Moreover, four out of six nonsense

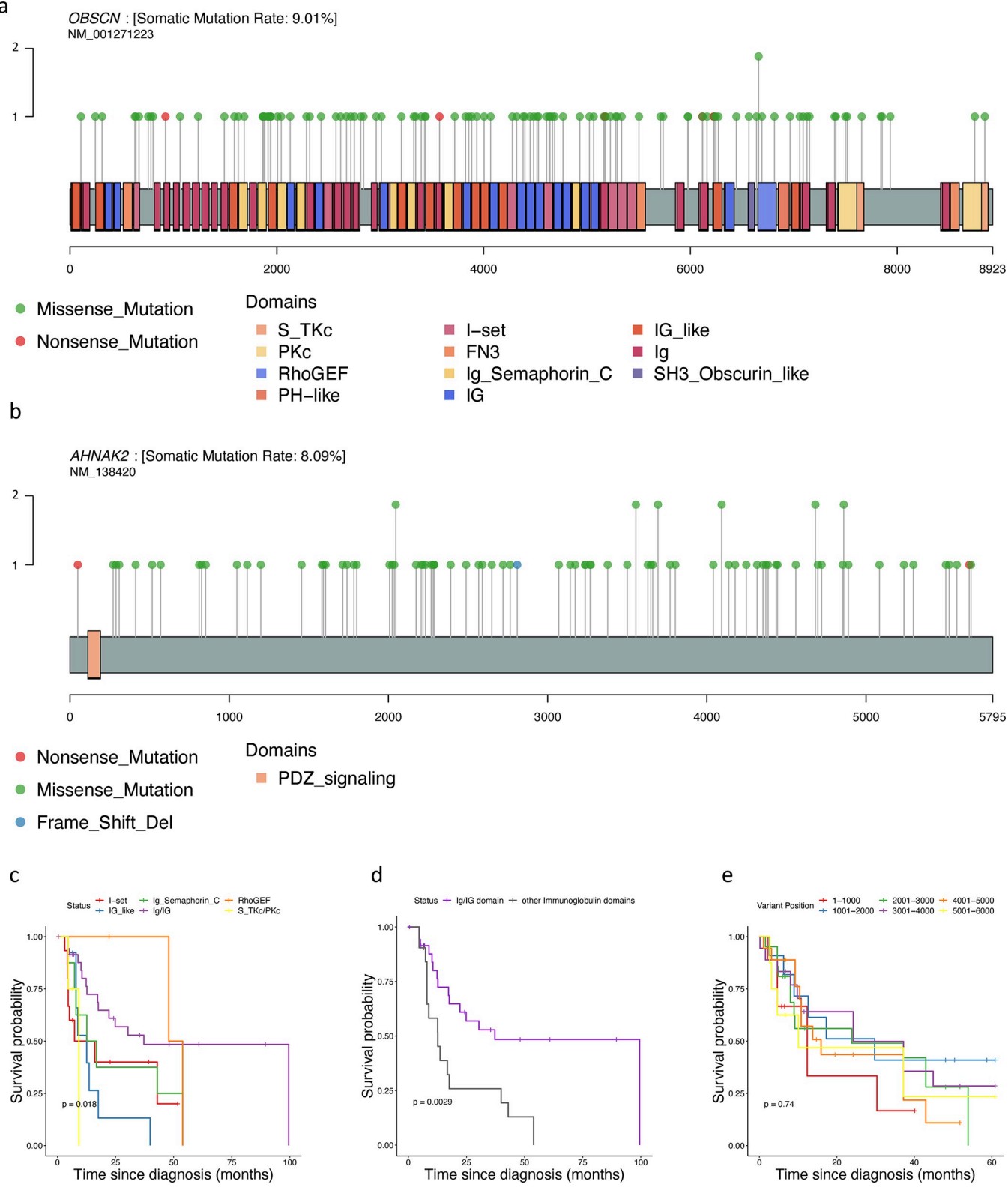

**Fig 5. Protein-level analysis of mutations of *OBSCN* and *AHNAK2* genes in GBM patients.** a) A lollipop plot illustrating various *OBSCN* mutations (missense or nonsense) in different domains of the OBSCN protein. b) A lollipop plot depicting various *AHNAK2* mutations (missense, nonsense, or frameshift deletions) throughout the AHNAK2 protein. c) The Kaplan-Meier curve for OS analysis between GBM patients with mutations in different domains of the OBSCN protein. d) The Kaplan-Meier curve for OS analysis in GBM patients with mutations in the immunoglobulin domain of the OBSCN protein

compared to patients with mutations in other immunoglobulin-related domains. e) The Kaplan-Meier curve for OS analysis in GBM patients with mutations in different segments of the AHNAK2 protein.

mutations were found in the Ig domain, which may result in loss of function and OBSCN protein disruption, leading to improved survival in GBM patients. Our analysis revealed only the PDZ-signaling domain for the AHNAK2 protein, and none of the GBM patients had mutations in this domain. However, we divided the AHNAK2 protein sequence into six segments, each containing 1000 amino acids, and evaluated the survival of patients with mutations in each segment. The Kaplan-Meier analysis showed no significant difference between the patients with mutations in different segments of the AHNAK2 protein (Fig 5E).

## Discussion

Glioblastoma multiforme remains a highly malignant type of brain tumor with a poor prognosis despite the current standard of care [3]. The identification of genetic markers that can predict the prognosis of GBM patients is crucial for developing more effective treatment strategies.

In this study, we analyzed somatic mutation data from WES of GBM patients from extensive public datasets to identify frequently occurring genetic alterations and their potential clinical implications. Our analysis identified ten common genes that were frequently mutated across all four GBM cohorts, including *PTEN*, *TP53*, *TTN*, *MUC16*, *FLG*, *PCLO*, *MUC17*, *HMCN1*, *AHNAK2*, and *OBSCN*. Patients with mutations in *OBSCN* or *AHNAK2* had a more favorable OS. Furthermore, co-mutation of OBSCN and *AHNAK2* was associated with longer OS and higher TMB compared to patients with mutations in either gene alone.

*OBSCN* is a large gene that codes for giant obscurin proteins, which are crucial for the organization and activity of muscle cells [15]. *OBSCN* has been identified to play a critical role in cancer and is highly mutated across different cancer types, including pancreatic and breast cancers, with a mutation frequency of 5–8% and 11.43%, respectively [16–18]. Furthermore, a study that compared the six-month progression-free survival between good and poor prognosis GBM patients revealed that six genes, including *OBSCN*, were significantly mutated in the poor prognosis group. This contradicts our findings, which suggest that *OBSCN* mutations are associated with favorable survival outcomes in GBM patients [19]. *OBSCN* undergoes several alternative splicing, resulting in multiple isoforms that range in size from 40 to 870 kDa. Although the small and intermediate obscurin remain largely unexplored, the giant obscurin, obscurin-A (~720 kDa), and obscurin-B (~870 kDa), have been extensively studied. These giant obscurins consist of tandem immunoglobulin and fibronectin-III domains, as well as several signaling motifs, including an IQ motif that binds to calmodulin and a tripartite cassette composed of Src homology 3 (SH3) motif, a RhoGEF motif, and a pleckstrin homology (PH) domain [20]. Our study has demonstrated that various mutations in distinct domains of *OBSCN* result in varying survival outcomes, and immunoglobulin domain mutations had better survival compared to other Ig domain mutations. Therefore, allele-level mutations of *OBSCN* may cause different clinical implications in different types of cancer.

AHNAK2 belongs to the cytoskeletal family of proteins called AHNAK that are involved in cell adhesion, migration, and signaling [21, 23]. AHNAK2 is over-expressed in several types of cancer, including clear cell renal cell carcinoma, pancreatic ductal adenocarcinoma, uveal melanoma, papillary thyroid carcinoma, and lung adenocarcinoma, where its high expression levels have been linked to poor patient prognosis [21–25]. A study conducted by Wang et al. demonstrated that the knockdown of AHNAK2 can inhibit cell proliferation, migration, and invasion while promoting apoptosis, indicating its potential oncogenic role in the progression

of lung adenocarcinoma [26]. Additionally, an analysis of the regulatory mechanisms that contribute to cancer development has revealed that subclonal mutations of *AHNAK* and *AHNAK2* in GBM can impact crucial molecules and processes linked to glioma progression [27]. Furthermore, in this study we analyzed drug sensitivity based on both *OBSCN* and *AHNAK2* mutational status and showed that OSI.906 and BMS.754807 have the potential to sensitize patients with double mutations in *OBSCN* and *AHNAK2* compared to other phenotypes.

While our study provides valuable insights into the potential use of genetic markers in predicting prognosis and guiding treatment decisions for GBM patients, some limitations should be taken into account. Firstly, we used data from four independent cohorts, which may have introduced certain biases and limitations. Additionally, our study only analyzed somatic mutations and did not consider other genetic alterations, such as copy number variations and epigenetic modifications, which also play a role in GBM development and progression. Furthermore, the impact of the tumor microenvironment and immune cells, particularly macrophages, on glioma progression and survival has not been considered in our analysis [28, 29]. Finally, we did not perform functional assays or in vivo models to validate the mutations of *OBSCN* or/and *AHNAK2* in GBM patients, which makes it difficult to ascertain their clinical implications and their association with favorable survival outcomes.

Future studies are needed to overcome these limitations and fully explore the potential of *OBSCN* or/and *AHNAK2* in the prognosis and treatment of GBM.

## Conclusion

In summary, the integration of multiple GBM datasets has revealed critical mutations in the GBM landscape. Our study introduces a novel classification based on the mutation status of *OBSCN* and *AHNAK2* among GBM patients. The distinct prognostic implications and molecular characteristics observed within the three OBSCN and *AHNAK2* mutant phenotypes provide valuable insights. Notably, the Double-Mut phenotype has led to the identification of potential antitumor drugs, showing promise for customized therapies precisely designed for these specific mutant phenotypes. Moreover, our exploration of protein-level mutations within *OBSCN* and *AHNAK2* has revealed the potential impact of specific protein domains on patient survival, suggesting that particular mutations within these domains could result in unique prognostic outcomes. Our study underscores the importance of precision medicine approaches in GBM treatment and highlights the potential of genetic marker analysis in improving patient outcomes. However, further investigation is needed to validate these findings and explore the therapeutic potential of *OBSCN* and *AHNAK2* as potential biomarkers and targets for GBM treatment.

## Supporting information

**S1 File. Contains all data for S1-S3 Tables and S1, S2 Figs.**
(DOCX)

## Acknowledgments

We would like to thank the Iranian National Brain Mapping Laboratory (NBML) and the National Institute for Medical Research Development (NIMAD) for their support throughout the research process.

## Author Contributions

**Formal analysis:** Parisa Azimi, Mina Karimpour.

**Investigation:** Taravat Yazdanian.

**Methodology:** Parisa Azimi.

**Resources:** Mehdi Totonchi.

**Software:** Parisa Azimi.

**Supervision:** Mehdi Totonchi, Abolhassan Ahmadiani.

**Visualization:** Abolhassan Ahmadiani.

**Writing – original draft:** Parisa Azimi.

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
