## [Decision Letter · Decision Letter 0]

8 Aug 2023

PONE-D-23-20005Comprehensive somatic mutational analysis in glioblastoma: implications for precision medicine approachesPLOS ONE

Dear Dr. Azimi,

Thank you for submitting your manuscript to PLOS ONE. After careful consideration, we feel that it has merit but does not fully meet PLOS ONE’s publication criteria as it currently stands. Therefore, we invite you to submit a revised version of the manuscript that addresses the points raised during the review process.

 Please address the objections brought forward by the reviewers, and adjust your manuscript accordingly. Please explain in detail, where this might not be possible.

We look forward to receiving your revised manuscript.

Kind regards,

Michael C Burger, M.D.

Academic Editor

PLOS ONE

6. We notice that your supplementary figures are uploaded with the file type 'Figure'. Please amend the file type to 'Supporting Information'. Please ensure that each Supporting Information file has a legend listed in the manuscript after the references list.

Reviewers' comments:

Reviewer's Responses to Questions

**Comments to the Author**

1. Is the manuscript technically sound, and do the data support the conclusions?

Reviewer #1: Yes

Reviewer #2: Partly

2. Has the statistical analysis been performed appropriately and rigorously? 

Reviewer #1: N/A

Reviewer #2: No

3. Have the authors made all data underlying the findings in their manuscript fully available?

Reviewer #1: Yes

Reviewer #2: Yes

4. Is the manuscript presented in an intelligible fashion and written in standard English?

Reviewer #1: Yes

Reviewer #2: Yes

5. Review Comments to the Author

Reviewer #1: This topic is very interesting, but some points must be revised. Look carefully at these:

- "We identified at least ten commonly mutated genes... associated with favorable survival outcomes in these patients." This part seems methods.

- "Therefore, it is essential to identify new molecular targets that are involved in cell growth and survival to develop more effective therapeutic approaches for the GBM [4]" - In addition it is important to add the concept of "survivorship bias in glioblastoma". Consider PMID: 37392899.

- "Notably, MUC17-MUC16, MUC17-PCLO, PCLO-MUC16, TTN-PCLO, and OBSCN-TTN interactions showed the strongest correlations " Are there other correlations between other genetic alterations?

- "While our study provides valuable insights into the potential.. some limitations should be taken into account". Among limitations authors should add that not only genetic mutations affect OS, but they also discuss the role of the tumor microenvironment (TME) and immuno cell. Consider both these recent papers: -- PMID: 37218976 -- PMID: 34471502

- "Our findings provide new insights into the molecular... for the development of molecular-targeted therapies." At the end of the manuscript it is not clear whether this article is a review or a clinical research. If this is clinical research, what do the authors add new compared to previous literature? Improve conclusion, what do they want to report new?

Reviewer #2: I appreciate the opportunity to review "Comprehensive somatic mutational analysis in glioblastoma: implications for precision medicine approaches" by Azimi et al. focusing on glioblastoma multiforme (GBM), a topic of significant importance in oncology. The authors have chosen a rigorous approach and clear presentation of the material.

However, following careful consideration and review, I regretfully cannot recommend acceptance of this paper in its current form. My main concern is related to the novelty and impact of the findings. According to the data and methods employed in the manuscript, it seems that the work represents a reiteration of previous findings. The literature contains several published works that have produced almost identical results, utilizing similar methodologies.

Moreover, the analysis employed, though robust in its design, could have been more comprehensive. Specifically, a systematic analysis including data from all available databases could have added more depth and relevance to the research. Instead, the study focused on a rather limited selection of the ten most common genes found in the specific datasets. This choice potentially limits the generalizability of the findings and may not fully represent the complexity of GBM.

Additionally, while the manuscript is well written and the data clearly presented, the conclusions drawn do not seem to be sufficiently supported by the data. For example, the identification of OBSCN and AHNAK2 mutations and their association with more favorable overall survival and specific drug sensitivity patterns seems very interesting; however, it is crucial that these observations are substantiated with more robust and specific data.

In summary, while the manuscript offers a worthwhile investigation of GBM, it would greatly benefit from further analyses to augment the novelty and impact of the findings. I believe that incorporating the above-mentioned considerations would significantly strengthen the manuscript, ultimately leading to a more rigorous and impactful contribution to the existing body of knowledge on GBM.

I hope that my feedback will be constructive for the authors and that the outlined concerns can be addressed in a revised version of the manuscript.

6. PLOS authors have the option to publish the peer review history of their article (what does this mean?). If published, this will include your full peer review and any attached files.

Reviewer #1: No

Reviewer #2: **Yes: **Georgios Markopoulos

---

## [Author Response · Author response to Decision Letter 0]

7 Nov 2023

5 September 2023

Dear Editorial Committee,

MS: No. PONE-D-23-20005

Comprehensive somatic mutational analysis in glioblastoma: implications for precision medicine approaches

Thank you for your e-mail. We found the reviewers’ comments very helpful. Please find the following point-by-point responses as requested:

Comments to the Author

1. Is the manuscript technically sound, and do the data support the conclusions?

Reviewer #1: Yes

Reviewer #2: Partly

2. Has the statistical analysis been performed appropriately and rigorously?

Reviewer #1: N/A

Reviewer #2: No

3. Have the authors made all data underlying the findings in their manuscript fully available?

Reviewer #1: Yes

Reviewer #2: Yes

4. Is the manuscript presented in an intelligible fashion and written in standard English? PLOS ONE does not copyedit accepted manuscripts, so the language in submitted articles must be clear, correct, and unambiguous. Any typographical or grammatical errors should be corrected at revision, so please note any specific errors here.

Reviewer #1: Yes

Reviewer #2: Yes

5. Review Comments to the Author

Reviewer #1:

This topic is very interesting, but some points must be revised. Look carefully at these:

- "We identified at least ten commonly mutated genes... associated with favorable survival outcomes in these patients." This part seems methods.

The following sentence “We identified at least ten commonly mutated genes in the investigated datasets including OBSCN and AHNAK2 genes. Upon integrating the mutational and clinical information, we discovered the co-mutation of OBSCN and AHNAK2 holds promise as a biomarker associated with favorable survival outcomes in these patients” was changed to “Our analysis revealed several commonly mutated genes, which could potentially indicate survival outcomes in GBM patients” in introduction section of new version as suggested.

- "Therefore, it is essential to identify new molecular targets that are involved in cell growth and survival to develop more effective therapeutic approaches for the GBM [4]" - In addition it is important to add the concept of "survivorship bias in glioblastoma". Consider PMID: 37392899.

Thank you for your comment. We have included the concept of "survivorship bias in glioblastoma" in the introduction section of the new version. “In addition, the development of prognostic GBM biomarkers often relies on surgically obtained tumor samples. However, bias due to varying surgeon selection criteria affects the accuracy of future analyses [12]. Thus, by integrating multiple datasets and considering larger sample sizes, it appears that the current limitations could potentially be mitigated.”

- "Notably, MUC17-MUC16, MUC17-PCLO, PCLO-MUC16, TTN-PCLO, and OBSCN-TTN interactions showed the strongest correlations" Are there other correlations between other genetic alterations?

Thank you for your inquiry. Apart from the mentioned correlations, we have provided additional significant interactions in new version of the Supplementary Table 2 for your reference.

- "While our study provides valuable insights into the potential. Some limitations should be taken into account". Among limitations authors should add that not only genetic mutations affect OS, but they also discuss the role of the tumor microenvironment (TME) and immune cell. Consider both these recent papers: -- PMID: 37218976 -- PMID: 34471502

Thank you for your valuable comments, we have added the suggested limitation into our discussion section. “Furthermore, the impact of the tumor microenvironment and immune cells, particularly macrophages, on glioma progression and survival has not been considered in our analysis [28-29]”

- "Our findings provide new insights into the molecular... for the development of molecular-targeted therapies." At the end of the manuscript, it is not clear whether this article is a review or clinical research. If this is clinical research, what do the authors add new compared to previous literature? Improve conclusion, what do they want to report new?

We have re-written the conclusion section of the new version file to emphasize the novel contributions of our study. The revised conclusion section is now more focused into GBM's molecular landscape and its relevance for developing targeted therapies. We believe these clarifications underscore the novelty and significance of our research.

The conclusion section was revised completely, as suggested by reviewers

“Conclusion: In summary, the integration of multiple GBM datasets has revealed critical mutations in the GBM landscape. Our study introduces a novel classification based on the mutation status of OBSCN and AHNAK2 among GBM patients. The distinct prognostic implications and molecular characteristics observed within the three OBSCN and AHNAK2 mutant phenotypes provide valuable insights. Notably, the Double-Mut phenotype has led to the identification of potential antitumor drugs, showing promise for customized therapies precisely designed for these specific mutant phenotypes. Moreover, our exploration of protein-level mutations within OBSCN and AHNAK2 has revealed the potential impact of specific protein domains on patient survival, suggesting that particular mutations within these domains could result in unique prognostic outcomes. Our study underscores the importance of precision medicine approaches in GBM treatment and highlights the potential of genetic marker analysis in improving patient outcomes. However, further investigation is needed to validate these findings and explore the therapeutic potential of OBSCN and AHNAK2 as potential biomarkers and targets for GBM treatment.”

Reviewer #2:

I appreciate the opportunity to review "Comprehensive somatic mutational analysis in glioblastoma: implications for precision medicine approaches" by Azimi et al. focusing on glioblastoma 

multiforme (GBM), a topic of significant importance in oncology. The authors have chosen a rigorous approach and clear presentation of the material.

However, following careful consideration and review, I regretfully cannot recommend acceptance of this paper in its current form. My main concern is related to the novelty and impact of the findings. According to the data and methods employed in the manuscript, it seems that the work represents a reiteration of previous findings. The literature contains several published works that have produced almost identical results, utilizing similar methodologies. Moreover, the analysis employed, though robust in its design, could have been more comprehensive. Specifically, a systematic analysis including data from all available databases could have added more depth and relevance to the research. Instead, the study focused on a rather limited selection of the ten most common genes found in the specific datasets. This choice potentially limits the generalizability of the findings and may not fully represent the complexity of GBM. 

In this study, we analyzed all available whole exome sequencing (WES) databases containing mutations of GBM patients, which include TCGA, CGGA, CPTAC and MAYO-PDX. The aim of this study was to find important genes that are mutated at a high rate in GBM patients using an unbiased bioinformatic data analysis. By our comprehensive bioinformatic data analysis, we have candidate a list of ten frequently mutated genes (PTEN, TP53, TTN, MUC16, FLG, PCLO, MUC17, HMCN1, AHNAK2, and OBSCN). Form our list, we found several novel candidate genes that is important to be discussed and considered in future. Among these ten genes, we selected those that had a significant relationship with overall survival and that had not been previously studies. However, we have prepared a supplementary table1 that shows all common mutated genes that highlighted by analyzing aforementioned four databases. In addition, we checked the overall survival for all these genes, which can be seen in the Supplementary Table 1.

Additionally, while the manuscript is well written and the data clearly presented, the conclusions drawn do not seem to be sufficiently supported by the data. For example, the identification of OBSCN and AHNAK2 mutations and their association with more favorable overall survival and specific drug sensitivity patterns seems very interesting; however, it is crucial that these observations are substantiated with more robust and specific data.

This section has re-written accordingly.

In summary, while the manuscript offers a worthwhile investigation of GBM, it would greatly benefit from further analyses to augment the novelty and impact of the findings. I believe that incorporating the above-mentioned considerations would significantly strengthen the manuscript, ultimately leading to a more rigorous and impactful contribution to the existing body of knowledge on GBM.

This was revised.

I hope that my feedback will be constructive for the authors and that the outlined concerns can be addressed in a revised version of the manuscript.

6. PLOS authors have the option to publish the peer review history of their article (what does this mean?). If published, this will include your full peer review and any attached files.

Do you want your identity to be public for this peer review? For information about this choice, including consent withdrawal, please see our Privacy Policy.

Reviewer #1: No

Reviewer #2: Yes: Georgios Markopoulos

I hope you find the corrections satisfactory.

Wish you all the best.

Yours sincerely

Azimi, P.

---

## [Decision Letter · Decision Letter 1]

28 Nov 2023

Comprehensive somatic mutational analysis in glioblastoma: implications for precision medicine approaches

PONE-D-23-20005R1

Dear Dr. Azimi,

We’re pleased to inform you that your manuscript has been judged scientifically suitable for publication and will be formally accepted for publication once it meets all outstanding technical requirements.

Kind regards,

Michael C Burger, M.D.

Academic Editor

PLOS ONE

Additional Editor Comments (optional):

Reviewers' comments:

Reviewer's Responses to Questions

**Comments to the Author**

1. If the authors have adequately addressed your comments raised in a previous round of review and you feel that this manuscript is now acceptable for publication, you may indicate that here to bypass the “Comments to the Author” section, enter your conflict of interest statement in the “Confidential to Editor” section, and submit your "Accept" recommendation.

Reviewer #1: All comments have been addressed

Reviewer #2: (No Response)

2. Is the manuscript technically sound, and do the data support the conclusions?

Reviewer #1: Yes

Reviewer #2: Yes

3. Has the statistical analysis been performed appropriately and rigorously? 

Reviewer #1: Yes

Reviewer #2: N/A

4. Have the authors made all data underlying the findings in their manuscript fully available?

Reviewer #1: Yes

Reviewer #2: Yes

5. Is the manuscript presented in an intelligible fashion and written in standard English?

Reviewer #1: Yes

Reviewer #2: Yes

6. Review Comments to the Author

Reviewer #1: The authors addressed all my criticisms. This paper provides important insights into the genetic alterations and somatic interactions in GBM, which could have implications for the development of new therapeutic strategies for this aggressive malignancy.

Reviewer #2: The resubmitted version of the manuscript provides a more comprehensive approach that includes additional explored genes. Based on this addition, I recommend acceptance.

I would also suggest a more detailed discussion of the additional findings, based on the current literature

7. PLOS authors have the option to publish the peer review history of their article (what does this mean?). If published, this will include your full peer review and any attached files.

Reviewer #1: No

Reviewer #2: **Yes: **Georgios Markopoulos

---

## [Editor Report · Acceptance letter]

20 Dec 2023

PONE-D-23-20005R1 

PLOS ONE

Dear Dr. Azimi, 

I'm pleased to inform you that your manuscript has been deemed suitable for publication in PLOS ONE. Congratulations! Your manuscript is now being handed over to our production team.

Kind regards, 

on behalf of

Dr. Michael C Burger 

Academic Editor

PLOS ONE